# *XYLT1* Deficiency of Human Mesenchymal Stem Cells: Impact on Osteogenic, Chondrogenic, and Adipogenic Differentiation

**DOI:** 10.3390/ijms26157363

**Published:** 2025-07-30

**Authors:** Thanh-Diep Ly, Vanessa Schmidt, Matthias Kühle, Kai Oliver Böker, Bastian Fischer, Cornelius Knabbe, Isabel Faust-Hinse

**Affiliations:** 1Institut für Laboratoriums- und Transfusionsmedizin, Herz- und Diabeteszentrum Nordrhein-Westfalen, Universitätsklinik der Ruhr-Universität Bochum, Medizinische Fakultät OWL (Universität Bielefeld), Georgstraße 11, 32545 Bad Oeynhausen, Germany; 2Department of Trauma Surgery, Orthopaedics and Plastic Surgery, University Medical Center Goettingen, Georg-August-University, 37075 Goettingen, Germany

**Keywords:** Xylosyltransferase-I (XT-I), human mesenchymal stem cells (hMSCs), chondrogenic differentiation, osteogenic differentiation, skeletal development, cartilage integrity, gene editing, bone homeostasis, CRISPR-Cas9

## Abstract

Xylosyltransferase-I (XT-I) plays a crucial role in skeletal development and cartilage integrity. An XT-I deficiency is linked to severe bone disorders, such as Desbuquois dysplasia type 2. While animal models have provided insights into XT-I’s role during skeletal development, its specific effects on adult bone homeostasis, particularly in human mesenchymal stem cell (hMSC) differentiation, remain unclear. This study investigates how XT-I deficiency impacts the differentiation of hMSCs into chondrocytes, osteoblasts, and adipocytes—key processes in bone formation and repair. The aim of this study was to elucidate for the first time the molecular mechanisms by which XT-I deficiency leads to impaired bone homeostasis. Using CRISPR-Cas9-mediated gene editing, we generated *XYLT1* knockdown (KD) hMSCs to assess their differentiation potential. Our findings revealed significant disruption in the chondrogenic differentiation in KD hMSCs, characterized by the altered expression of regulatory factors and extracellular matrix components, suggesting premature chondrocyte hypertrophy. Despite the presence of perilipin-coated lipid droplets in the adipogenic pathway, the overall leptin mRNA and protein expression was reduced in KD hMSCs, indicating a compromised lipid metabolism. Conversely, osteogenic differentiation was largely unaffected, with KD and wild-type hMSCs exhibiting comparable mineralization processes, indicating that critical aspects of osteogenesis were preserved despite the *XYLT1* deficiency. In summary, these results underscore XT-I’s pivotal role in regulating differentiation pathways within the bone marrow niche, influencing cellular functions critical for skeletal health. A deeper insight into bone biology may pave the way for the development of innovative therapeutic approaches to improve bone health and treat skeletal disorders.

## 1. Introduction

Skeletal dysplasias (SD) encompass a heterogeneous group of genetic conditions defined by abnormalities in bone and cartilage growth, development, and maintenance. These conditions often result in irregularly shaped bones, particularly affecting the head, spine, and long bones of arms and legs. Despite their individual rarity, SDs collectively have a significant prevalence, affecting approximately one in every 3000 to 5000 live births. Many of these disorders arise from mutations in genes essential for the biosynthesis, structure, and regulation of the extracellular matrix (ECM), underscoring the critical role of ECM components in maintaining skeletal integrity [1,2,3]. The ECM, comprising components such as collagens, glycosaminoglycans (GAG), and proteoglycans (PG), forms complex networks providing mechanical support and anchoring sites for cells and growth factors [4].

Desbuquois SD type 2 (DBQD2, MIM 608,124) is one such SD, characterized by facial deformities, growth retardation, and short long bones. This rare autosomal recessive disorder results from mutations in the *XYLT1* gene, encoding the enzyme xylosyltransferase-I (XT-I; EC 2.4.2.26) [5]. The latter plays a crucial role in GAG synthesis by catalyzing the initial and rate-limiting step of transferring xylose to specific serine residues on PG core proteins, thereby initiating the assembly of GAG chains, such as chondroitin sulfate and heparan sulfate [6]. The PGs, including aggrecan (ACAN) and perlecan (HSPG2), are fundamental ECM components that contribute to cartilage resilience, bone strength, and overall tissue homeostasis [7]. Xylosyltransferase-II, an isoform of XT-I encoded by the *XYLT2* gene (Gene ID: 64132), also contributes to GAG synthesis with similar catalytic functions. The *XYLT2* mutations are associated with spondylo-ocular syndrome (OMIM # 605822) [8]. Although *Xylt1* and *Xylt2* are similarly expressed in adult mice, *Xylt2* knockout mice develop postnatal liver and kidney cysts but no skeletal defects [9]. This highlights the distinct but overlapping roles these isoenzymes play in different tissues and the critical importance of their functions in maintaining ECM integrity.

Bone homeostasis involves the dynamic and continuous remodeling of bone tissue maintained by the coordinated activities of osteoblasts and osteoclasts to balance bone formation and bone resorption. Membrane-bound or soluble forms of the receptor activator of nuclear factor kappa-Β ligand (RANKL), primarily expressed by stromal cells and osteoblasts, regulate osteoclastogenesis in the bone microenvironment. Osteoprotegerin (OPG) modulates the half-life of membrane-bound RANKL, and, interestingly, GAGs inhibit the OPG-induced shortening of RANKL’s half-life, thus influencing the balance of bone remodeling. In addition, bone mineralization and remodeling are closely linked and influence the structural properties of bone. Human mesenchymal stem cells (hMSC) are central to bone homeostasis, as they differentiate into osteoblasts, chondrocytes, and adipocytes. However, hMSCs are also able to differentiate into other cell types, such as, for instance, neurons. The bone-associated differentiation is regulated by key transcription factors, such as SRY-box 9 (SOX9), runt-related transcription factor 2 (RUNX2), and peroxisome proliferator-activated receptor gamma (PPARG). The SOX9 is essential for maintaining the chondrocyte lineage, while RUNX2 promotes chondrocyte hypertrophy and osteoblast differentiation, and PPARG affects both osteogenesis and adipogenesis. Dysregulation of these cell types, which share a common mesenchymal progenitor lineage, can significantly impact bone formation, bone marrow composition, and the balance between fat and bone [7,10,11,12,13,14].

A critical aspect of bone health is its regenerative capacity, particularly evident in fracture healing. Chondrocytes are the cells responsible for cartilage formation, playing a crucial role in endochondral ossification, which is essential for bone development. Chondrocytes also play a critical role in fracture repair by mimicking skeletal development. Fracture repair involves complex processes to restore bone to its original form, characterized by two primary pathways: intramembranous and endochondral ossification. Stem or progenitor cells differentiate directly into osteoblasts in intramembranous bone formation, leading to the deposition of mineralized bone matrix. This pathway is observed predominantly in stable fractures. Osteoblastogenesis consists of three key stages: proliferation, matrix maturation, and mineralization. During these stages, specific markers and enzymes are expressed. Alkaline phosphatase (ALP) and collagen type I alpha 1 (COL1A1) indicate the beginning of osteoblast differentiation early in the process. As the process continues, osteocalcin, which is encoded by *BGLAP*, becomes important, signaling the mineralization stage. Osteopontin (OPN), encoded by the *SPP1* gene, shows increased activity twice: once during proliferation and again in the later stages. In the final stage, mature osteoblasts produce ALP, OPN, and osteocalcin and are found next to newly formed bone tissue. By contrast, endochondral bone formation, common in unstable fractures or those with large defects, involves the differentiation of stem/progenitor cells into chondrocytes, forming a cartilaginous matrix later replaced by bone. The transition from cartilage to bone necessitates the differentiation of proliferating chondrocytes into hypertrophic chondrocytes, characterized by the high expression of RUNX2 and low expression of cartilage-specific markers, such as collagen type II alpha 1 (COL2A1) and SOX9. Subsequently, the hypertrophic chondrocytes synthesize an ECM that has a different composition than that of proliferating cartilage, mediating their transdifferentiation to osteoblasts [13,15,16,17,18]. In the context of osteoarthritis (OA), the process of chondrocyte hypertrophy is thought to contribute to the degradation of COL2A1, highlighting the susceptibility of the ECM to pathological changes [16]. Furthermore, the serum XT-I level increases during early posttraumatic OA in mice with high bone-forming potential, suggesting broader implications for this gene in cartilage-related disorders [19]. Despite advances in understanding XT-I’s role during skeletal development through animal models [20,21], its specific role in adult bone homeostasis, particularly regarding MSC differentiation, remains unclear. The hMSCs from bone marrow are key players in bone homeostasis and serve as an optimal in vitro model for studying XT-I deficiency due to their differentiation potential into osteoblasts, chondrocytes, and adipocytes [22,23].

Using CRISPR-Cas9-mediated gene editing to generate *XYLT1*-deficient (knockdown, KD) hMSCs, we intend to analyze the molecular characteristics and functions of KD hMSCs and their derived cell types to elucidate the impact of impaired PG biosynthesis on adult skeletal integrity and metabolism. The results of this study will improve our understanding of the biological mechanisms underlying bone homeostasis and pave the way for innovative therapeutic approaches to treat skeletal disorders associated with XT-I deficiency.

## 2. Results

### 2.1. Successful Generation of XYLT1-Deficient hMSCs by CRISPR-Cas9 Genome Editing

We employed CRISPR-Cas9 gene editing to generate KD hMSCs to investigate the role of *XYLT1* in hMSC differentiation and bone homeostasis. We utilized a predesigned gRNA specifically targeting the coding region of exon 3 in the *XYLT1* gene, facilitating the introduction of insertions or deletions (indels) to disrupt the *XYLT1* expression in hMSCs [24]. Following transfection, fluorescence microscopy revealed a high proportion of hMSCs exhibiting strong fluorescence signals compared to untreated control cultures, indicating efficient RNP complex uptake (Figure A1A). The quantification of transfection efficiency using flow cytometry analysis demonstrated that 98% of hMSCs were successfully transfected (Figure A1B,C), confirming the effective delivery of the RNP complex into the cells. We performed Sanger sequencing (Figure A2A) to verify that the CRISPR-Cas9 RNP complex achieved effective genome editing, and a T7EI assay (Figure A2B) to detect potential indel formations at the target site. These validations confirmed the presence of mutations at the target site, indicating successful editing. Further confirmation of the indel formation was provided by TA cloning, and the sequencing analysis of individual clones revealed heterogeneous indels, including deletions and insertions (Figure A2C), thereby confirming the gene editing of the *XYLT1* exon 3 coding region. Following the validation of on-target effects, potential off-target effects were examined. The two genomic DNA sequences identified by the *XYLT1* gRNA manufacturer as the most probable off-target sites were sequenced to rule out off-target impacts. Analysis of the corresponding regions within the *LMO3* and *SLC9A2* genes confirmed that the use of the *XYLT1* gRNA did not result in any mutations in these areas (Figure A3).

Following the transfection and validation of the effectiveness of the CRISPR-Cas9 strategy by FACS (Figure A1), genotyping, and sequencing (Figure A2), showing successful *XYLT1* gene editing in hMSCs cultures, qRT-PCR analyses and enzymatic XT-I activity assay were performed to evaluate the impact of CRISPR-Cas9 editing on the target gene expression and functional enzyme level (Figure 1).

Quantitative qRT-PCR analysis at the mRNA level demonstrated a significant reduction in target gene expression by 64% compared to mock-transfected controls (Figure 1A). In addition, the mRNA expression of the xylosyltransferase-II isoform *XYLT2* was unaffected upon *XYLT1* targeting (Figure 1B). The measurement of cellular XT-I activity showed a corresponding decrease in target protein activity (Figure 1C), indicating a successful target gene KD at both the transcriptional and functional level in hMSCs.

In summary, the efficient generation of *XYLT1*-deficient hMSC primary cultures provides a reliable and foundational tool for investigating the impact of XT-I deficiency on cellular processes critical for bone and cartilage development.

The entire experimental procedure in each subsequent experiment was conducted in accordance with standard protocols, incorporating the quantification of transfection efficiency (Appendix A), Sanger sequencing (Appendix A), TA cloning (Appendix A), and off-target verification (Appendix A), with the objective of ensuring consistent initial conditions across all experimental setups. Furthermore, the expression levels of *XYLT1* mRNA and XT-I activity were evaluated independently for each experimental condition (Figure A4, Figure A5 and Figure A6). This approach was adopted to validate the reliability and specificity of the CRISPR-Cas9-mediated gene editing process and ascertain the subsequent differentiation outcome.

### 2.2. XYLT1 Deficiency Does Not Impact the Cellular Proliferation and Senescence of hMSCs

Cellular senescence impacts the differentiation potential of hMSCs and, thus, the generation and number of different cell types, such as, for instance, vital bone cells [25]. Therefore, the impact of *XYLT1* deficiency on the proliferation and senescence of hMSCs was assessed by several in vitro assays comparing *XYLT1*-deficient hMSCs and WT controls (Figure 2).

In order to assess whether *XYLT1* deficiency affects the proliferation and senescence of hMSCs, the cell proliferation was evaluated using the WST-1 assay, revealing no significant difference in cell growth rates between *XYLT1*-deficient and control cells (Figure 2A). Additionally, the SA-β-gal assay showed comparable levels of SA-β-gal activity in both groups, indicating that diminished *XYLT1* expression does not induce premature cellular senescence (Figure 2B). Furthermore, the expression levels of key cell cycle regulators, p21 (*CDKN1A1*) and p53 (*TP53*), involved in premature senescence [26], were similar in WT and KD hMSCs, as confirmed by qRT-PCR (Figure 2C,D). Interestingly, the relative expression of ECM-associated PG genes in hMSCs was unaffected upon CRISPR-Cas9-mediated *XYLT1* KD, and the expression of neither integrin alpha-5 (*ITGA5*) nor transforming growth factor β1 (*TGFB1*) was regulated (Appendix A).

Together, these findings suggest that *XYLT1* deficiency does not adversely affect the proliferative capacity or senescence of hMSCs.

### 2.3. Diminished XYLT1 mRNA Expression of hMSCs Affects Their Chondrogenic Differentiation Potential

Cartilage-to-bone transition in fracture healing is highly dependent on hypertrophic chondrocytes that become osteoprogenitors and osteoblasts [15]; therefore, we validated the chondrogenic differentiation process in our cell culture model. The validation of chondrogenic differentiation in verified WT and KD hMSC cultures (Figure A4) was performed by gene expression analysis (Figure 3) of early and late chondrogenic differentiation markers and histological staining (Figure 4 and Figure 5). Due to the substantial number of cells needed for the chondrogenic spheroid culture, the undifferentiated control condition was omitted from the gene expression analysis. It was found in the analysis of pellet-cultured hMSCs undergoing chondrogenic differentiation that *XYLT1* deficiency resulted in a notable 30% decrease in *SOX9* expression. The expression of *COL2A1* decreased by 50%, while *ACAN* expression increased by 1.3-fold (Figure 3B,C). The *RUNX2* expression increased by 1.6-fold, and *COL1A1* expression was reduced by 40% (Figure 3D,E). Lastly, *MMP13* expression was 20% lower in the absence of *XYLT1* (Figure 3F). It is important to highlight that the *ACAN* gene expression results were the only ones that showed a complete contrast between individual experiments concerning male vs. female hMSC pools, with some experiments indicating suppression and others showing an induction of *ACAN* expression in KD hMSC-derived chondrocyte cultures compared to WT (Appendix A).

Following the analysis of early gene expression at the 48 h time point, which provides insights into the initial molecular changes during the chondrogenic differentiation of WT and KD hMSCs, the study was extended to examine changes at the cellular and structural levels. Cultures were stained with alcian blue (Figure 4) and Sirius red (Figure 5) on days 14 and 28 to visualize the long-term effects of differentiation and evaluate the development of the ECM. These stains allow for the assessment of PG-rich structures, mainly consisting of ACAN and collagen fiber formation [27], providing a comprehensive view of the differentiation processes over time.

Alcian blue staining is used to detect acidic PG and glycoproteins, which are key components of the ECM, particularly in cartilage tissue. The dye binds to sulfated and carboxylated groups within the PG GAG side chains, resulting in a turquoise blue coloration.

The analysis of alcian blue-stained cryosections from spheroids composed of either WT or KD hMSCs revealed significant differences in the chondrogenic differentiation (Figure 4). The differentiation procedure was conducted twice, using three biological and three technical replicates for each experiment. All samples in cultures maintained in standard medium exhibited only dark blue to purple nuclear staining at both day 14 and day 28, indicating the absence of chondrogenic differentiation due to the lack of differentiation-inducing factors. The WT hMSCs displayed pronounced turquoise blue staining at day 14 in the first chondrogenic differentiation experiment, reflecting active chondrogenic differentiation and substantial PG synthesis. By contrast, KD hMSCs showed no turquoise blue staining, suggesting impaired differentiation at this stage. The intensity of the turquoise blue staining in WT hMSCs decreased by day 28 compared to day 14, possibly due to matrix remodeling processes. Conversely, KD hMSCs exhibited increased turquoise blue staining at day 28 compared to day 14, indicating a delayed but eventual improvement in differentiation, reaching levels similar to WT. The WT hMSCs in the second differentiation experiment showed only slight turquoise blue staining at day 14, while KD hMSCs demonstrated more intense staining, suggesting that, under these conditions, KD cells might possess an enhanced differentiation potential. The WT hMSCs exhibited intense turquoise blue staining by day 28, indicating robust matrix formation. The KD hMSCs also showed turquoise blue staining, but to a lesser extent than the WT, indicating ongoing differentiation with some limitations. The similarity between the KD staining at day 28 in this experiment and the WT staining at day 28 from the first experiment suggests possible matrix remodeling or adaptation in KD cells.

These findings indicate that *XYLT1* deficiency affects the chondrogenic differentiation of hMSCs, resulting in variable outcomes. The WT hMSCs generally demonstrated consistent and timely chondrogenic differentiation. By contrast, KD hMSCs show indications of either delayed or prematurely initiated chondrogenic differentiation. This highlights the complex role of *XYLT1* in regulating the timing and extent of chondrogenesis of hMSCs.

The Alizarin Red staining of cryosections from WT and KD hMSCs revealed notable differences in the formation of collagenous fibers (Figure 5). Cells cultured in standard medium did not show red-orange-stained collagenous fibers; instead, the cryosections were dominated by brown-black nuclear staining, indicating a lack of ECM development under these conditions. The WT sections displayed red staining between the gray nuclei on day 14 in the first differentiation experiment, indicating the presence of collagenous fibers. The KD sections also exhibited red staining, but to a lesser extent compared to WT, suggesting reduced fiber formation. The WT sections showed an increase in red fiber staining by day 28, indicating a further development of collagenous fibers compared to day 14. The KD sections also showed an increased red fiber staining compared to their day 14 counterparts, but the extent of fiber formation was still less than that observed in WT sections at day 28. The WT sections exhibited red-stained collagenous fibers on day 14 in the second differentiation experiment, while KD sections showed red staining that was less distributed and more locally concentrated. The staining in KD sections appeared more intense than in WT in some areas, but the fibers were less structured. The WT sections demonstrated a marked increase in staining and collagenous fiber formation by day 28, indicating robust matrix development. Sections using differentiated KD hMSCs also showed an increase in red staining compared to their undifferentiated controls, with fibers appearing thinner and less organized compared to sections derived from the differentiated WT cells.

These observations suggest that *XYLT1* deficiency influences the structure and distribution of collagenous fibers during the differentiation process. The KD hMSCs tend to develop reduced fiber formation with fibers that are less organized and thinner than those in WT hMSCs, indicating a potential impact on the overall maturation and structural integrity of the ECM.

### 2.4. Diminished XYLT1 Expression of hMSCs Does Not Critically Impair Their Differentiation into Osteoblasts

We cultured both WT and KD hMSCs under osteogenic conditions to determine whether *XYLT1* expression is necessary for the osteogenic differentiation of hMSCs. The basal expression levels of *XYLT1* mRNA and XT-I activity were assessed to ensure a diminished XT-I level across all experimental KD setups (Figure A5), thereby validating the reliability of the differentiation outcomes observed. Interestingly, while the mRNA expression level of *XYLT1* in both WT and KD hMSCs remained unchanged during osteogenic differentiation compared to undifferentiated controls, an increase in the expression level of *XYLT2* was observed at the 14-day differentiation time point (Figure A5A,B).

The differentiation of hMSCs into mature osteoblasts was comprehensively assessed using both gene expression analysis and biochemical assays (Figure 6 and Figure 7).

The expression of key osteogenic markers, including *RUNX2*, *SOX9*, *OPG*, *COL1A1*, *BGLAP*, and *SPP1,* was quantified via qRT-PCR (Figure 6). The *RUNX2* mRNA expression showed no significant differences between WT and KD hMSCs after 7 and 14 days of cultivation in either standard or osteogenic differentiation medium (Figure 6A). On day 7, the *SOX9* mRNA expression increased 1.4-fold in differentiated WT hMSCs, but this increase was absent in differentiated KD hMSC cultures compared to undifferentiated controls. By day 14, no significant differences in the *SOX9* expression were observed in any hMSC cultures (Figure 6B). The *OPG* mRNA expression increased significantly after 7 and 14 days in differentiation medium, with 12.9-fold and 7.3-fold rises in WT hMSCs, and 12.1-fold and 8.0-fold increases in KD hMSCs, respectively (Figure 6C). The *COL1A1* mRNA expression was induced by day 14 in osteogenic medium, with a 1.7-fold increase in WT and a 2.1-fold increase in KD hMSCs compared to controls (Figure 6D). On day 7, *BGLAP* mRNA expression decreased by 70% in both WT and KD hMSCs compared to controls. By day 14, *BGLAP* expression was 40% lower in KD hMSCs, though not significantly different in differentiated cells (Figure 6E). The *SPP1* mRNA expression decreased on day 7 in both WT and KD hMSCs, returning to baseline by day 14. The *SPP1* expression was initially 40% lower in KD hMSCs, a reduction that persisted after 14 days in both media types (Figure 6F). In addition, the relative expression of ECM-associated PG genes *DCN*, *SDC2*, and *TGFB1* in osteogenically differentiated hMSCs was unaffected upon CRISPR-Cas9-mediated *XYLT1* KD, except for a slight induction of *DCN* expression on day 7. Furthermore, the relative expression quantification of *ITGA5*, which has been shown to be involved in dexamethasone-induced osteoblast differentiation [28], was not affected by *XYLT1* deficiency either (Appendix A). These results suggest that *XYLT1*-deficient hMSCs exhibit similar capabilities to WT cells in terms of the expression of osteogenesis key markers and PG core proteins, but exhibit differences regarding the gene expression of mineralization modulators.

Biochemical assays addressing the ECM mineralization were performed to obtain a more detailed understanding of the mineralization status during osteogenic differentiation and confirm that the gene expression changes observed are translated into functional and structural characteristics typical of bone tissue development. Early differentiation stages were monitored by assessing the ALP activity, which showed no significant variation between the groups (Figure 7A). Inorganic phosphate measurements further corroborated these findings, showing comparable mineral content at both day 7 and day 14 (Figure 7B). The visualization of the mineralized matrix of osteogenically differentiated WT and KD hMSCs was performed by Alizarin Red S staining. The evaluation of the mineralization capacity revealed similar levels of calcium deposition in both WT and *XYLT1-*deficient hMSCs at day 14 (Appendix A), but fewer calcified nodules appeared in a bright red color at day 7 in KD compared to WT cultures of one differentiation procedure performed (Figure 7C). The OPN protein levels determined by ELISA, in a similar fashion to the *SSP1* expression level quantified (Figure 6F), showed no significant differences between differentiated WT and KD hMSCs at both time points, but a significant reduction in undifferentiated cells at the 7-day time point (Figure 7D). These results collectively suggest that the initiation of the osteogenic differentiation process is preserved in KD hMSCs, indicating that the XT-I activity within the PG synthesis pathway, though diminished, does not critically impair the cellular mechanisms driving osteogenesis.

### 2.5. XYLT1-Deficient Adipogenically Differentiated hMSCs Possess a Diminished Leptin Expression

The balance between osteoblast and adipocyte formation within the bone marrow environment is essential for maintaining healthy bone density and function. Adipocytes within the bone marrow can influence bone metabolism through the secretion of various factors that affect osteoblast and osteoclast activity [29,30]. Therefore, studying the adipogenic differentiation in hMSCs, particularly in the context of *XYLT1* deficiency, is crucial for elucidating the molecular mechanisms that regulate bone homeostasis. Adipogenic differentiation was conducted using verified WT and KD hMSC primary cultures in alignment with the differentiation experiments performed. The *XYLT1* deficiency was confirmed by sequencing exon 3 of the *XYLT1* gene (Appendix A), TA cloning (Appendix A), off-target (Appendix A) verifications, and gene expression analysis (Figure A6). The qRT-PCR showed a significant reduction of *XYLT1* mRNA expression in RNP-transfected cells, while *XYLT2* expression remained unchanged. This reduction persisted throughout the 20-day differentiation, resulting in decreased XT-I enzyme activity in KD compared to WT hMSCs.

Subsequently, we examined the protein expression of perilipin 1 and leptin (Figure 8), both crucial in mature adipocytes, with perilipin coating the surface of intracellular lipid droplets [31] and leptin functioning as an adipokine [32] with osteogenic modulating capacities [30,33]. Both control and differentiated WT and KD hMSC cultures were analyzed on day 20 of adipogenic differentiation by immunofluorescence microscopy for perilipin 1 protein quantification. Perilipin was detected in hMSC cultures using a specific antibody for perilipin 1, while the adipocyte lipid droplets were visualized using BODIPY staining (Figure 8A). The analysis demonstrated successful adipogenic differentiation in both WT and KD hMSC cultures, as evidenced by the presence of visible lipid droplet structures—surrounded by perilipin 1 and stained with BODIPY—in primary cell cultures treated with differentiation media. These organelle structures were absent in the undifferentiated control cultures of both WT and KD hMSCs. Notably, undifferentiated control KD hMSC cultures on day 20 were more likely to exhibit round BODIPY-positive organelle structures, despite the absence of detectable perilipin 1 staining in either WT or KD controls. Regarding the adipogenically differentiated hMSCs, WT cultures were found to have a visually greater amount of BODIPY-positive lipid droplets compared to KD cultures. Perilipin 1 protein expression was quantified using ImageJ software by calculating the normalized CTCF from images corresponding to each experimental condition. When comparing perilipin 1 expression per cell in adipogenically differentiated WT and KD hMSC cultures, no significant differences were found (Figure 8B).

Since leptin is secreted primarily by functional adipocytes, with its levels determined largely by the number of adipocytes present [32] as well as the size of the lipid droplets [34,35], we used an ELISA to quantify the leptin expression in cell lysates from adipogenically differentiated WT and KD hMSC cultures. This approach allowed us to estimate the size and number of lipid droplets, as the overall perilipin expression in WT and KD hMSCs did not differ. The results showed that the amount of leptin protein was 40% lower in the cell lysates of KD hMSCs compared to those of WT hMSC cultures, consistent with the amount of BODIPY-stained lipid droplets in the hMSC cultures observed (Figure 8C). Given that larger adipocytes possess higher intracellular leptin concentrations than smaller cells [35], this suggests potential differences in the cell and lipid droplet size between WT and KD adipocyte cultures.

We conducted initial gene expression analyses to evaluate the impact of *XYLT1* deficiency on adipogenic differentiation to reinforce our initial findings and understand the molecular mechanisms involved better (Appendix A). Among the genes evaluated in this single study iteration, *PPARG*, a crucial transcription factor for adipocyte differentiation [36], showed nonsignificant differences between WT and KD hMSC cultures (Appendix A). By contrast, the gene encoding perilipin 1, *PLIN1*, exhibited a significant reduction to 70% upon *XYLT1* deficiency (Appendix A). Consistent with the protein level determination, the expression of the gene encoding leptin, *LEP*, was significantly reduced by 60% in KD hMSCs compared to WT (Appendix A). Furthermore, the gene encoding heparan sulfate PG perlecan, *HSPG2*, associated with ECM structure and adipocyte cell size [37], showed a nonsignificant suppression to 60% (Appendix A). Although these findings are preliminary and require further validation, they offer a molecular context to the protein expression patterns observed, suggesting that *XYLT1* deficiency may influence specific aspects of adipocyte gene expression and function.

## 3. Discussion

The XT-I deficiency has been linked to severe skeletal abnormalities, such as those observed in DBQD2 [5,38,39,40,41] or Baratela–Scott syndrome [42,43], highlighting the enzyme’s importance in skeletal development and cartilage integrity. Previous research projects studying *XYLT1* deficiency in bone-related pathologies used different animal models, such as mouse [21,44] and zebra fish larvae [20]. Focusing on the role of *XYLT1* deficiency during skeletal development using animal models helps us to understand the cellular events leading to the onset of rare human bone disorders. However, the specific effects of XT-I deficiency on adult bone homeostasis remain poorly understood, especially concerning its impact on hMSC differentiation, which is fundamental to bone formation during skeletal homeostasis and bone repair.

Here we generated KD hMSCs by CRISPR-Cas9-mediated gene editing to examine their differentiation potential and analyze how XT-I deficiency impacts cellular mechanisms. Understanding the factors that influence the onset of these skeletal disorders will enable better diagnosis and prognosis, as well as the development of personalized treatment options.

We generated hMSC cultures with the desired gene KD without the use of plasmids for CRISPR editing, consistent with previous works using RNP-mediated CRISPR-Cas9 genome editing of hMSCs [45]. The KD efficiency achieved in this study was comparable with previous works that utilized the RNP- or nanoparticle-based approaches [45,46]. As hMSCs experience a significant reduction in differentiation capacity with repeated in vitro expansion [47], typical single-cell clonal expansion within the experimental workflow could not be realized to prevent in vitro hMSC aging. Hence, the presence of some WT or heterozygous cells in the population is unavoidable. We examined the KD cell proliferation capacity and markers of cellular senescence to neglect the overgrowth of individual WT cells within this heterogeneous primary KD cell population. These factors, which could also lead to growth arrest and the reduced differentiation potential of hMSCs [25], were found to be comparable to those in WT cultures.

Our findings contrast with previous studies on KD primary human fibroblast cultures generated by CRISPR-Cas9 gene editing, which reported a notable induction of cellular senescence linked to *XYLT1* deficiency [24,48]. Although primary fibroblasts share morphological characteristics and some cell surface markers with hMSCs [49,50], they are terminally differentiated cells with limited regenerative capacity. This suggests that fibroblasts may be more vulnerable to senescence in response to KD-mediated ECM changes compared to hMSCs, probably due to their specialized functions and reduced adaptability [51]. Thus, hMSCs demonstrate greater resilience to ECM alterations, which is attributed to their inherent plasticity and regenerative potential [52]. This assumption is supported by our experimental data, showing marginal changes in the expression levels of ECM-associated genes, previously shown to be markedly regulated upon *XYLT1* suppression in different cell types [48,53].

Based on these findings, we focused on the chondrogenic differentiation of KD and WT hMSCs into chondrocytes, a process significantly influenced by *XYLT1* during bone development [20,21]. We observed an increase in the *RUNX2* and *ACAN* expression in KD hMSC cultures, alongside reduced levels of *SOX9*, *COL2A1*, *COL1A1*, and *MMP13* in our analysis of the chondrogenic differentiation at the early differentiation time point of 48 h—characterized by induced *XYLT1* expression in chondrogenically differentiated hMSCs [54]. These data suggest that diminished *XYLT1* mRNA expression and enzyme activity impact important regulatory factors, as well as the expression of abundant cartilage ECM components. Notably, the downregulation of *SOX9* and the upregulation of *RUNX2* observed here indicate that proliferative chondrocytes are entering hypertrophy [55]. The downregulation of *SOX9* at the 48 h time point in our experimental setup aligns with a reduction in the expression of its downstream target gene, *COL2A1*. As a master transcription factor, *SOX9* is crucial for chondrogenesis, sequentially regulating a series of downstream factors in a stage-specific manner [56].

In addition, we also observed aberrant collagen and PG expressions in KD hMSC-derived chondrocyte cultures compared to WT by histological staining at later differentiation time points, indicating that the diminished XT-I expression in hMSCs affects the timing and proper maturation of functional chondrocytes and, thus, the ECM morphology of the cartilage. These differences were especially observed for the timing of the chondrocyte marker *ACAN* expression in both the qRT-PCR and histological staining experiments performed. This finding is supported by results from XylT-I/cKO mice, in which tibia sections with alcian blue staining of PGs show significantly reduced GAG content in resting and proliferative zones, but not in hypertrophic zones [57].

Notably, there was a significant reduction in the hypertrophic region in XylT-I/cKO mice compared to control mice, supporting the idea that the hypertrophic zone is diminished in the mutants. This reduction may result from a greater proportion of cells transitioning from the hypertrophic state to terminal differentiation and ossification [57]. Our findings further reinforce this hypothesis. Given that hypertrophic cartilage serves as a transitional tissue between cartilage and bone [55], we observed a notable remodeling of the ECM in KD hMSC cultures. This remodeling is characterized by a transition from an ACAN- to a collagen-rich ECM between days 14 and 28 of chondrogenic differentiation. The shift in gene expression and ECM composition suggests a potential alteration in the differentiation pathway of KD hMSCs, underscoring the necessity for further investigation into the underlying regulatory mechanisms.

Individuals with DBQD2 often suffer from OA [58,59,60]. Understanding chondrocyte hypertrophy is crucial due to its role in OA [61]. Research shows that mice lacking one copy of the Runx2 gene have reduced cartilage degradation [62], indicating that mediators such as XT-I might affect disease progression. Levels of XT-I and ACAN are reduced in OA, and XT-I expression exhibits periodic oscillation in chondrocytes [63]. Collectively, these findings indicate that chondrogenically differentiated KD hMSCs progress more rapidly toward terminal differentiation or ossification. Our data support and extend previous observations by Taieb et al. [57], confirming that proper XT-I expression is essential to prevent chondrocyte hypertrophy. This aligns with earlier studies indicating that human XT-I expression is induced in the early stages of chondrogenic differentiation of hMSCs [21,54] and is absent in hypertrophic chondrocytes of murine WT bone tissue [57]. Therefore, our findings indicate that XT-I plays a crucial role in regulating susceptibility to cartilage degeneration by preventing premature chondrocyte hypertrophy and maintaining PG homeostasis.

Although chondrogenic differentiation is compromised due to the reduced *XYLT1* expression in KD hMSCs, our findings demonstrate that osteogenic differentiation remains effectively sustained. This conclusion is reinforced by the comparable ALP activity and inorganic phosphate levels observed in both KD and WT hMSCs. While the expression of late osteoblast regulatory genes is indeed reduced, the overall mineralization process and the expression of early osteogenic markers remain intact. This suggests that KD hMSCs may retain critical aspects of osteogenic differentiation, potentially through compensatory mechanisms that favor mineral deposition. Additionally, the structural analysis of trabecular bone from WT and KD mice revealed significantly increased mineral deposition associated with *XYLT1* deficiency [57]. Our study observed a notable reduction in the expression of the mineralization inhibitor SPP1 in osteoblast cultures derived from KD hMSCs compared to WT, in line with findings from murine Opn^−/−^ osteoblasts, which showed increased mineral deposition compared to their WT counterparts without significant changes in the osteogenic marker expression [64]. This decrease in the *SPP1* expression highlights the complex role of OPN/SPP1 in promoting mineralization during osteoblastic differentiation, which may contribute to the overall higher bone mineral density observed with *XYLT1* deficiency. In addition, the interaction between OPN and integrin αv/β1 is crucial for the lineage determination of MSCs [64]. We did not observe any significant differences in the *ITGA5* expression between osteogenically differentiated WT and KD hMSCs in our study. However, we noted a reduced expression of *SPP1*, suggesting that the adipogenic cell fate is more favored in KD hMSC cultures compared to WT. Previous research indicates that intracellular lipid droplets can enhance the osteoblast function [65] and that inhibiting OPN function promotes adipogenic differentiation in MSCs [66]. These observations suggest a compensatory mechanism in KD hMSCs, indicating a shift toward adipogenic differentiation. Recent studies show that XylT2 is crucial for lipid homeostasis, as its deficiency leads to adipose tissue loss and lipodystrophy in mice [67]. As such, the increase in *XYLT2* mRNA at the 14-day mark in osteogenically differentiated hMSC cultures may help mitigate the loss of *XYLT1* expression.

It is well-established that adipogenesis, osteogenesis, and osteoclastogenesis are interconnected processes. Thus, an imbalance in adipogenesis is associated with various human metabolic diseases, including osteoporosis [29]. Therefore, we explored the differentiation of WT and KD hMSCs to adipocytes to understand better how *XYLT1* deficiency impacts adipogenesis. Our findings revealed a successful adipogenic differentiation in both WT and KD hMSC cultures, marked by the presence of perilipin 1-coated lipid droplets [31]. While the overall perilipin expression per cell remained unchanged in KD cultures, the total *PLIN* expression was diminished. This apparent discrepancy can be attributed to the role of perilipins, which are lipid droplet surface proteins that stabilize and protect lipid droplets from lipase action. The transcriptional regulation of perilipin by PPARγ and its posttranslational modification via proteasomal degradation are key factors influencing perilipin levels; therefore, the reduction in overall *PLIN* expression observed could be due to a lack of lipolysis, preventing the degradation of lipid droplets and leading to an accumulation of PLIN1. This accumulation may trigger a feedback mechanism that reduces the *PLIN1* gene expression, as suggested by previous findings. Consistent with this, our gene expression analysis reveals that the key adipogenic transcription factor *PPARG* shows no significant difference between WT and KD hMSC-derived adipocyte cultures. This indicates that the initial stages of adipogenesis may remain intact despite any *XYLT1* deficiency. However, we observed a nonsignificant reduction in the *HSPG2* expression to 60%, suggesting potential changes in the cellular structural integrity and function [37]. Wilsie et al. demonstrated, using the 3T3-L1 adipogenic model, that cell surface HSPGs are vital for adipocyte differentiation from pre-adipocytes [68]. They found that adipogenic induction increases GAG biosynthesis, whereas treatment with xylosides, potent XT inhibitors [69], led to reduced GAG chain assembly and lipid uptake [68]. The impaired *HSPG2* expression in KD hMSCs suggests a disruption in the regulatory balance necessary for effective lipid droplet formation in adipocytes. Regarding lipid metabolism, the expression of *LEP*, which encodes leptin, is significantly reduced by 60% in KD hMSCs, consistent with the corresponding protein data. This reduction implies a smaller adipocyte cell size and lower lipid content in KD hMSC-derived adipocyte cultures compared to WT, as supported by previous studies [70,71]. Although the *PPARG* expression remains stable, it is insufficient alone to maintain normal *LEP* expression, highlighting the need for additional regulatory factors to control leptin levels, such as *HSPG2* [72,73]. Furthermore, leptin plays a crucial role in modulating the suppressive effects of PPARγ on chondrogenic differentiation and T3-mediated chondrocyte hypertrophy [74], indicating that proper adipocyte function may have wider effects on maintaining the cellular functions of other cell types within the bone environment. Overall, these findings highlight the complex interplay of molecular factors required for successful adipogenesis and underscore the pivotal role of XT in maintaining this balance within the bone marrow niche.

The study’s limitations, such as the reliance on monoculture models and the use of dexamethasone, which may alter differentiation outcomes by suppressing osteocalcin and failing to induce RUNX2 [75,76], highlight the need for a careful interpretation of the findings. Looking forward, future research should address those limitations by employing coculture systems to mimic the in vivo bone environment better and explore the dynamic interactions between various cell types involved in bone homeostasis. Incorporating single-cell analyses could offer more detailed insights into the differentiation potential and heterogeneity of hMSCs. Additionally, increasing the sample size with more primary cell-derived hMSCs and examining sex-dependent differences will enhance the robustness and applicability of the findings.

## 4. Materials and Methods

### 4.1. Cell Culture

Four hMSC primary cultures from bone marrow were obtained from PromoCell (Heidelberg, Germany) in passage two and expanded, according to the manufacturer’s instructions. After their initial expansion, ensuring comparable cell morphology and growth rate of all primary cultures, in each case, two donor-derived hMSC primary cultures were dissociated, counted, and combined equally in cell number for cryopreservation and later usage in the gene editing experiments. We thus generated two cell pools consisting of equal proportions of two primary cell cultures. Pools #1 and #2 were used in all experimental setups. This procedure should mitigate the individual biological variability and improve the reproducibility of the data obtained. The general donor information of the initial primary cells used in this study for generating the hMSC cultures is listed in Table 1.

The hMSCs (5 × 10^3^ cells/cm^2^) were maintained under standardized culture conditions (37 °C, 5% CO_2_) in 0.2 mL hMSC growth medium (MSC Growth Medium 2; PromoCell, or MSCGM; Lonza, Basel, Switzerland) per cm^2^ of tissue culture surface area after thawing the passage 3 cultures. Medium renewal was performed every two to three days. Cells were subcultured by reaching 90% confluency using a solution of 0.05% (*w*/*v*) trypsin and 0.02% (*w*/*v*) EDTA in Dulbecco’s phosphate-buffered saline (DPBS; Thermo Fisher Scientific, San Diego, CA, USA) for cell detachment.

Regarding the osteogenic differentiation of hMSCs, 1 × 10^4^ cells/cm^2^ were maintained in 0.2 mL hMSC growth medium (MSC Growth Medium 2; PromoCell) per cm^2^ of tissue culture surface area to obtain a cell monolayer at 100% confluency within 72 h. Duplicate samples were prepared for all experimental setups and time points given. One of the duplicate samples was induced with differentiation medium (MSC Osteogenic Differentiation Medium; PromoCell), while the other was maintained in growth medium as a negative control for 7 or 14 days, respectively.

Regarding the chondrogenic differentiation of hMSCs, the hMSC Chondrogenic Differentiation Medium BulletKit^TM^ (Lonza), which consists of an incomplete chondrogenic induction medium (control) and a complete chondrogenic induction medium supplemented with TGF-β_3_ (differentiation), was used. The chondrogenic pellet cultures were prepared, according to the manufacturer’s instructions. Briefly, a total of 2.5 × 10^5^ hMSCs was needed to form each chondrogenic pellet. Duplicate pellet samples were prepared for all experimental setups and time points given. The amount of hMSCs required to form the number of pellet cultures needed was obtained and transferred into a centrifuge tube for washing with 1 mL incomplete chondrogenic medium per 7.5 × 10^5^ hMSCs. The resulting cell pellet was resuspended in 1 mL incomplete chondrogenic medium per 5.0 × 10^5^ hMSCs. A suspension volume corresponding to 2.5 × 10^5^ hMSCs was transferred into polypropylene tubes and centrifuged (150× *g*, 5 min). After discarding the supernatants, the resulting cell pellets were each resuspended in 0.5 mL of control or chondrogenic differentiation medium. After centrifugation, the resulting cell pellets were maintained under standardized culture conditions with the tube caps loosened to allow gas exchange and cultured for a total of 14 or 28 days.

The hMSC Adipogenic Differentiation Medium BulletKit^TM^ (Lonza) was utilized, which consists of an induction and a maintenance medium, for the adipogenic differentiation of hMSCs. In brief, 2.1 × 10^4^ cells/cm^2^ were maintained in 0.2 mL growth medium (MSCGM^TM^, Lonza) per cm^2^ of tissue culture surface area. Duplicate samples were prepared for all experimental setups and time points given. Once 100% confluency was reached, one of the duplicate samples was subjected to three cultivation cycles consisting of a 72 h cell cultivation period in induction medium (hMSC Adipogenic Induction Medium; Lonza) followed by a 24 h cultivation period in maintenance medium (hMSC Adipogenic Maintenance Medium; Lonza), while the other duplicate sample was cultured solely in maintenance medium as a negative control on the same schedule. The adipogenic cultivation procedure was completed with an additional 7-day culture period in maintenance medium.

### 4.2. Ribonucleoprotein-Based CRISPR-Cas9-Mediated XYLT1 KD in hMSCs

The CRISPR-Cas9-mediated gene editing in hMSCs presented in this work utilized the Alt-R^TM^ CRISPR-Cas9 System and solutions from Integrated DNA Technologies, Inc. (IDT; Coralville, IA, USA) and the Lipofectamine^TM^ CRISPRMAX^TM^ Cas9 Transfection Reagent (CRISPRMAX; Thermo Fisher Scientific Inc.). The crRNA:tracrRNA complex was hybridized beforehand and combined with Cas9, resulting in the ribonucleoprotein (RNP) complex. A *XYLT1*-specific predesigned Alt-R CRISPR-Cas9 crRNA, previously utilized [24], and a fluorescently labeled ATTO^TM^ 550 Alt-R^®^ CRISPR-Cas9 tracrRNA were each resuspended in Nuclease-Free IDTE Buffer (10 mM Tris, 0.1 mM EDTA; IDT) to 100 µM stock concentrations. In order to create a final gRNA concentration of 1 µM, crRNA and tracrRNA were mixed equimolarly and diluted in Nuclease-Free Duplex Buffer (30 mM HEPES, pH 7.5; 100 mM potassium acetate; IDT) and incubated for 2 min at 95 °C. The gRNA oligos generated were allowed to cool to room temperature (RT, 20–25 °C) in the dark. The RNP complex was formed by combining 11.5 µL of the gRNA solution (1 µM), 23.0 µL of Alt-R^TM^ S.p. Cas9 Nuclease V3 (Cas9, 1 µM) diluted in Opti-MEM, 7.5 µL Cas9 Plus^TM^ Reagent from the CRISPRMAX kit (Thermo Fisher Scientific Inc.), and 458.0 µL Opti-MEM. After an incubation period of 5 min at RT, the RNP complex was combined with a transfection mixture, consisting of 9.2 µL CRISPRMAX and 490.8 µL Opti-MEM, and incubated for 20 min at RT in the dark. hMSCs were diluted to 24,000 cells per mL in antibiotic-free growth medium. An amount of 1 mL of the transfection solution was added to the wells of a 6-well tissue culture plate (Greiner Bio-One, Kremsmünster, Austria) before adding 2 mL of the prepared cell suspension for a final volume of 48,000 hMSCs per well for reverse transfection. The cells were incubated for 24 h, washed twice with DPBS, and maintained in standard growth medium. The transfection efficiency was visually determined 24 h post-transfection by fluorescent microscopy using the BZ-X810 microscope (Keyence, Osaka, Japan).

### 4.3. Fluorescence-Activated Cell Sorting

The use of the ATTO 550 fluorescent dye-labeled tracrRNA for RNP complex formation enabled the fluorescence-activated cell sorting (FACS) analysis for the enrichment of successfully transfected hMSCs. The ATTO 550 conjugate has an excitation peak at 553 nm and an emission peak at 575 nm. The FACS of hMSCs was performed 24 h post-transfection for the best flow cytometric resolution as the fluorescence intensity gradually decreases over time [77]. The cell monolayer was washed with DPBS and enzymatically detached from the tissue culture plate for the FACS of hMSCs. The cells recovered were resuspended in 500 µL growth media and subjected to FACS using the S3e Cell Sorter (BioRad Laboratories, Hercules, CA, USA). Cell suspensions of cells without lipofection and transfected cells without RNP complex were utilized as background controls and used to set the gates during FACS. A single-parameter histogram with the relative fluorescence or light scatter intensity (FL2, λ_ex/em_ = 553/575) on the *x*-axis and the number of events (cell counts) on the *y*-axis was used to determine the autofluorescence of the control cell populations and set the intensity cut-off for the positive and negative dataset. The positive and negative cell populations of the cell suspensions analyzed were separately enriched and cultured (5 × 10^3^ cells/cm^2^) in hMSC growth medium until 90% confluence and used for subsequent experiments.

### 4.4. Genomic DNA Amplification

In order to evaluate the genome editing of the *XYLT1* exon 3 region, the target region was amplified via polymerase chain reaction (PCR) using the HotStarTaq^®^ DNA Polymerase kit (Qiagen, Hilden, Germany). The reaction volume (25 µL) contained 11.15 μL distilled water, 2.5 μL 10× PCR buffer, 5 μL 5× Q-Solution^®^, 0.25 μL dNTPs (10 mM of each), 0.5 μL forward and reverse primer each (25 μM; Biomers, Ulm, Germany), 0.1 μL HotStarTaq DNA Polymerase (5 U/µL), and 5 μL DNA template (50 to 100 ng/µL). The PCR reaction was performed in a thermal cycler starting with an initial heat-activation step for 15 min at 95 °C. The three-step cycling program consisted of 35 cycles of denaturation (1 min, 95 °C), annealing (1 min, T_A_ of primer system), and elongation (1 min, 72 °C). The final extension was performed for 10 min at 72 °C. The purification of the PCR products generated was performed with the MSB^®^ Spin PCRapace kit (Invitek Molecular GmbH, Berlin, Germany), according to the manufacturer’s instructions.

### 4.5. Determining Genome Targeting Efficiency Using T7 Endonuclease I

The determination of genome targeting efficiencies was performed using a T7 Endonuclease I (T7EI) assay from NEB (Ipswich, MA, USA). Three PCR reactions were prepared for each amplicon setup: one containing the gDNA from targeted cells (RNP-transfected cells), another containing gDNA from negative control cells (mock-transfected cells), and a third one using water as a no-template control. Each annealing reaction contained 200 ng PCR product, mixed with 2 µL 10× NEBuffer 2, adding up to 19 µL with nuclease-free water. The reaction was heated to 95 °C (10 min) and then gradually cooled to 25 °C at a rate of −0.3 °C per second to hybridize the PCR products. A volume of 1 µL enzyme was added to the annealed PCR products and incubated for 15 min at 37 °C for T7E1 digestion. The reaction was stopped by adding 1.5 µL EDTA (0.25 M). The fragmented PCR products were visualized via 1.8% (*w*/*v*) agarose gel electrophoresis using the pUC19 DNA/MspI (HpaII) marker (Thermo Fisher Scientific Inc.). The relative quantification of the fragments post T7E1 digestion was performed with ImageJ 1.8.0 (National Institutes of Health, Bethesda, MD, USA).

### 4.6. TA Cloning for Single Allele Sequencing

A TA cloning was performed for the identification of target gene editing on the level of the single alleles via DNA sequencing. The Invitrogen TA Cloning^TM^ kit with a pCR^TM^ 2.1 Vector (Thermo Fisher Scientific Inc.) was utilized for the stochastic insertion of single PCR products into linearized plasmid vectors. Briefly, the ligation reaction contained PCR product (150 ng), 2 µL 5× T4 DNA Ligase Reaction Buffer, 2 µL pCR^®^ 2.1 vector (25 ng/µL), and water added to a final volume of 9 µL. After the addition of 1 µL ExpressLink^TM^ T4 DNA Ligase (5 U), the ligation reaction was incubated at 4 °C for 16 h and used to transform the construct into chemically competent TOP10 *E. coli*. A vial of competent cells (50 µL) was thawed on ice for 30 min and supplemented with 5 µL ligation reaction for each transformation. Cells were heat shocked for 35 s at 42 °C and incubated for 1.5 min on ice before the addition of 500 µL prewarmed S.O.C. medium (37 °C; Thermo Fisher Scientific Inc.). The vials were agitated for 30 min at 37 °C. A volume of 250 µL of each transformation vial was evenly distributed on a lysogeny broth agar plate containing 75 µg/mL ampicillin. The plates were incubated for 16 h at 37 °C. At least ten single colonies were selected from the plate and grown in 5 mL lysogeny broth liquid medium containing 75 µg/mL ampicillin for 16 h at 37 °C with constant agitation for plasmid isolation and restriction analysis. A volume of 2 mL bacteria cell suspension was centrifuged (6800× *g*) and processed with the QIAprep Spin Miniprep kit (Qiagen), according to the manufacturer’s instructions, for plasmid isolation. The purified plasmid DNA was obtained in 50 µL buffer EB and analyzed by sequencing for the presence of a single PCR insert to identify the single allele variants.

### 4.7. Sanger Sequencing

The DNA sequencing was performed using the BigDye^TM^ Terminator v3.1 Cycle Sequencing kit (Thermo Fisher Scientific Inc.). The Sanger sequencing reaction consisted of 3 µL of the purified PCR product, 2 µL BigDye™ Terminator 3.1 Ready Reaction Mix, 1.5 µL forward or reverse primer (2.5 µM), 4 µL BigDye™ Terminator v1.1 and v3.1 Sequencing Buffer (5×), and water added to a volume of 20 µL. The sequencing reaction started with a denaturation step at 95 °C for 120 s followed by 30 cycles of denaturation (95 °C, 10 s), annealing (T_A_, 10 s), and elongation (60 °C, 240 s). The eluate containing the fluorescently labeled PCR fragments was filtrated using Machery-Nagel^TM^ Receiver Columns (35 µm; Machery-Nagel, Düren, Germany) packed with 750 µL Sephadex^TM^ G-50 Superfine gel suspension prior to proceeding to automated sample loading of the ABI Prism^®^ 3500 Genetic Analyzer (Thermo Fisher Scientific Inc.) instrument for PCR product analysis via capillary electrophoresis. The electropherogram was analyzed using ABI PRISM^®^ Sequence Analysis Software v3.7.

### 4.8. Nucleic Acid Extraction, Quantification, and Synthesis of Complementary DNA

The extraction of RNA from cell lysates was conducted with the NucleoSpin^®^ RNA II kit (Machery-Nagel), according to the manufacturer’s instructions, except for the adipocyte and chondrocyte cultures. The Nucleospin Blood Kit (Machery-Nagel) was utilized for DNA extraction from cell lysates, according to the manufacturer’s instructions, for all primary cell cultures except the adipocyte cultures. The isolation of RNA and DNA from adipocyte cultures was performed with the AllPrep DNA/RNA Mini Kit (Qiagen), according to the manufacturer’s instructions. The RNA extraction and purification from the chondrocyte pellet lysate was carried out with the QIAzol Lysis Reagent and the RNeasy Mini Kit (Qiagen). The chondrocyte lysates were generated by using two cell pellets that were submerged in 700 µL QIAzol Lysis Reagent (Qiagen) and subjected to a repeated freeze–thawing cycle. The lysates were transferred into MP Biomedicals^TM^ Lysing Matrix S 2 mL tubes (MP Biomedicals, Santa Ana, CA, USA) and homogenized (7000× *g*, 15 s) using the MagNA Lyser (Roche, Basel, Switzerland). After a rest at RT for 5 min, the homogenate was transferred into a new tube with 140 µL chloroform and mixed for 15 s. The homogenate was left to stand (3 min, RT) and centrifuged (12,000× *g*, 15 min, 4 °C). The aqueous phase (~350 µL) was mixed with 100% ethanol (~525 µL), transferred to the RNeasy Mini Spin Column, and centrifuged (11,000× *g*, 15 s). The subsequent total RNA purification from chondrocyte pellets with the RNeasy Mini Kit (Qiagen) was performed according to the manufacturer’s instructions. The concentration and purity of the RNA extracted were determined with a spectrophotometer. The synthesis of complementary DNA using the isolated RNA as a template was carried out with the SuperScript II Reverse Transcriptase Kit (Thermo Fisher Scientific Inc.) and diluted 1:10 prior to qRT-PCR analysis.

### 4.9. Quantitative Real-Time PCR Analysis

The mRNA expression analysis was conducted, as previously described [78], via quantitative real-time PCR (qRT-PCR) using the LightCycler^®^ 480 SYBR Green I Master reaction mix for PCR, the LightCycler^®^ 480 Instrument II System (Roche) for real-time amplicon detection, and the respective primer systems (Table 2).

Expression data normalization was performed using the geometric mean of three reference gene expressions, including *GAPDH*, *RPL13A*, *B2M*, *YWHAZ*, or *HMBS*. All normalized gene expression data shown were referred to the target gene expression of the controls set to 1 for the relative comparison of multiple biological samples per experiment.

### 4.10. Bicinchoninic Acid Assay (BCA)

Protein concentrations were determined using the Pierce BCA Protein Assay Kit (Thermo Fisher Scientific Inc.), which is based on the reduction of Cu^2+^ to Cu^+^ by the sample proteins and the colorimetric detection of Cu^+^ by BCA. The BCA-Cu^+^ complex exhibits a linear absorbance at 562 nm with increasing sample protein concentrations. The assay procedure was carried out as described previously [79].

### 4.11. Mass Spectrometric XT-I Selective Activity Assay

The determination of XT-I activity in cell lysates and supernatants was performed using the ultra-performance liquid chromatography/electrospray ionization tandem mass spectrometry-based activity assay, as described previously [79].

### 4.12. Cell Proliferation Assay

The assay is based on the cleavage of the tetrazolium salt WST-1 to form a solubilized formazan product by metabolically active cells due to their mitochondrial dehydrogenase activities that can be spectrophotometrically quantified. The assay was performed in a 96-well microplate format with 1700 cells per well. Cells were cultured for 44 h in 100 µL culture medium under standard culture conditions before conducting the assay by supplementing the cell culture supernatant of each well with 10 µL WST-1 reagent. The spectrometric quantification of the formazan product was performed at a wavelength of 450 nm and a reference wavelength of 690 nm using a microplate reader. The measurements were performed directly after WST-1 supplementation and repeated thereafter every hour for the next 4 h.

### 4.13. Quantitative Determination of the Senescence-Associated β-Galactosidase (SA-β-gal) Activity

SA-β-gal activity is widely used to determine the replicative senescence in mammalian cells [80]. The enzymatic SA-β-gal activity can be quantified using the chromogenic substrate 5-bromo-4-chloro-3-indolyl-β-d-galactopyranoside (X-gal) that is converted to 7-hydroxy-4-methylcoumarin, which can be spectrophotometrically detected. The assay was conducted using cell lysates that were prepared beforehand by culturing cells at a density of 50 cells/cm^2^ in 6-well or 60 mm cell culture plates. The cell monolayer was lysed after washing three times with DPBS, using 150 µL or 300 µL lysis buffer, prepared as described previously [81]. The lysates of two wells of a six-well plate were combined, incubated (10 min, 4 °C), and stored at −80 °C for at least 24 h. The thawed lysates were centrifuged (5 min, 13,000× *g*, 4 °C), and the corresponding supernatants were used for the SA-β-gal activity assay.

### 4.14. Cryosection Procedure

The high-density pellet cultures were subjected to formaldehyde-based fixation (24 h, ROTI^®^ Histofix, Carl Roth GmbH + Co. KG, Karlsruhe, Germany), before freezing and sectioning was performed, to preserve the tissue morphology. In order to achieve a better visualization of the embedded tissue sample, the cell pellet was incubated in sucrose solution (15% (*w*/*v*) in PBS, 6 h) and transferred into tissue-freezing medium (Leica Biosystems Nussloch GmbH, Wetzlar, Germany) with methyl green (1:1 tissue-freezing medium and methyl green solution [0.8% (*w*/*v*) in PBS]). The pre-stained pellet tissue was completely embedded in tissue-freezing medium (Leica Biosystems), and frozen with dry ice and liquid nitrogen (−20 to −80 °C). The frozen embedded block specimens were stored at −80 °C until sectioning. The cryosections (5 µm) were obtained with the Leica CM3050 S research cryostat (Leica Biosystems) and mounted on frosted microscope slides (Thermo Fisher Scientific Inc.). The cryostat sections were stored at −20 °C and thawed (10 min, RT) before staining.

### 4.15. Alcian Blue Staining

The PG content of pellet-cultured chondrocyte cryosections was determined by alcian blue staining. The cationic alcian blue dye binds to highly negatively charged cellular structures, such as the sulfated GAG, via electrostatic interactions and enables their visualization [82]. Alcian blue staining solution (1% (*w*/*v*) in acetic acid [3% (*v*/*v*), pH 2.5]) was purchased from Morphisto GmbH (Frankfurt am Main, Germany). The staining procedure includes a cryosection incubation in water (10 s) and an incubation in filtered (0.22 µm syringe filter) alcian blue staining solution (5 min). The section was washed (3 min) with tap water and nuclear stained (1 min) in Gill’s III hematoxylin solution (Morphisto). After rinsing (3 min) in warm running tap water, the specimen was mounted between the microscope slide and coverslip using melted (55 °C) Kaiser’s glycerol gelatin (Carl Roth) as an aqueous coverslip sealant.

### 4.16. Picro-Sirius Red Staining

The staining of collagens within tissue sections by Picro-Sirius red was carried out using the Picro-Sirius red staining kit (Morphisto). The staining procedure includes a cryosection incubation in water (10 s) and an incubation (15 min) in Weigert’s iron hematoxylin solution. After washing (30 s) in distilled water, the sections were rinsed (8 min) in warm running tap water. After washing (1 min) in distilled water, the cryosections were stained (1 h) in Picro-Sirius red solution. After rinsing twice in acetic acid solution (3% (*v*/*v*)), the cryosections were rinsed (10 s) in distilled water and mounted using the melted (55 °C) aqueous coverslip sealant Kaiser’s glycerol gelatin (Carl Roth).

### 4.17. Alkaline Phosphatase Activity Assay in Cell Lysates

The colorimetric Alkaline Phosphatase Assay Kit (ab83369, Abcam, Cambridge, UK) was utilized for analysis, according to the manufacturer’s instructions. The cell lysates were prepared by mechanical cell lysis using a Dounce homogenizer and a total of 15 passes of the pestle in the cylindrical glass tube. The lysate supernatants were obtained after centrifugation (18,000× *g*, 4 °C, 15 min) and used for the ALP activity assay in predetermined dilutions. The assay is based on the spectrometric quantification of the yellow *para*-nitrophenol product formed by an enzymatic hydrolysis reaction using the colorless *para*-nitrophenylphosphate as a substrate reagent. The OD at 405 nm of the *para*-nitrophenol formed at pH 9.5 during a predetermined time period was determined with a microplate reader and is directly proportional to the sample’s ALP activity. The ALP activity (U/mL) in the test sample was calculated according to Equation (1), considering the amount *B* (in μmol) of *para*-nitrophenol in the sample well calculated from the standard curve, the reaction time *T* (in min), the original sample volume *V* (in mL) added into the reaction well, and the sample dilution factor *D*.(1)$$ALP\;activity=\left(\frac{B}{T\cdot V}\right)\cdot D$$

### 4.18. Quantification of Inorganic Phosphate from Cell Culture Supernatants

The concentration of inorganic phosphates in the cell culture supernatants was determined with the ARCHITECT c8000 clinical analyzer (Abbott Laboratories, Chicago, IL, USA). This determination method is based on the ultraviolet absorbance change that occurs in the reaction of inorganic phosphate with ammonium molybdate in the presence of sulfuric acid, forming a nonreduced phosphomolybdate that can be detected at 340 nm.

### 4.19. Enzyme-Linked Immunosorbent Assay

Quantitative determination of OPN and leptin concentrations in the cell culture supernatants or cell lysates was performed using their respective enzyme-linked immunosorbent assay (ELISA) kits. The Human Leptin Quantikine ELISA Kit (DLP00, R&D Systems™, Minneapolis, MN, USA) and the Human Osteopontin SimpleStep ELISA^®^ Kit (ab269374, Abcam) were employed, following the manufacturer’s instructions of each. The samples were stored at −80 °C, thawed, centrifuged (1000× *g*, 20 min, 4 °C), and diluted before usage for the respective ELISA. The color development after substrate addition was stopped, and its intensity during the respective assay period was spectrophotometrically determined with a microplate reader (Tecan Reader Infinity PRO, Tecan Group AG, Männedorf, Switzerland) set to the respective wavelengths. Wavelength correction was performed using the wavelength 570 nm that is subtracted from the optical density (OD) determined of each well at 450 nm. The resulting calculation was performed by averaging the duplicate reading results for each standard, control, and sample and subtracting the average zero standard OD. Creating a standard curve and performing a regression analysis gave the unknown protein concentration of the diluted sample.

### 4.20. Alizarin Red Staining

The Alizarin Red dye staining solution (40 mM in distilled water; Sigma-Aldrich, St. Louis, MO, USA) was used to identify calcium deposits in osteogenically differentiated hMSC cultures (48-well format). After a cultivation period of 7 or 14 days, the cell culture medium was removed from the wells, and the remaining cell monolayers were washed twice with DPBS. After adding 250 µL ROTI^®^ Histofix solution (4% (*v*/*v*) formaldehyde, phosphate-buffered, pH 7; Carl Roth) per well and fixation (10 min), the cells were washed twice with distilled water. Alizarin Red staining was performed by incubating (10 min) the cell monolayer in 250 µL filtered staining solution (pH 4.1) per well. The cell monolayer was washed five times in distilled water for complete dye removal. Microscopic analysis was performed by light microscopy.

### 4.21. Immunostaining and BODIPY Staining for Fluorescence Microscopy

Cells were cultured on eight-chamber slides prior to immunostaining. Every step in the immunostaining procedure described was followed by washing three times with DPBS. An incubation (30 min, 4 °C) with paraformaldehyde solution (500 µL) was performed for the fixation of the cell monolayer. Cells were permeabilized (10 min, RT) with a Triton X-100 solution (0.1% (*v*/*v*) in DPBS) to stain the intracellular proteins. The cell monolayer was incubated (1 h, RT) with a blocking solution (5% (*w*/*v*) bovine serum albumin in DPBS) for the blockage of unspecific binding sites. The fixed and permeabilized cell monolayer was incubated with a BODYPI solution (5 µM) for 1 h at RT in the dark and subsequently washed three times with DPBS before proceeding to the incubation step with the blocking solution. After blockage, the cell monolayer was incubated (1 h, RT, or 16 h, 4 °C) with a primary Anti-perilipin-1 (Abcam; ab3526) antibody solution (200 µL) diluted (1:400) in 1% (*w*/*v*) bovine serum albumin in DPBS, followed by a treatment (1 h, RT, darkness) with a secondary Alexa Fluor 555-conjugated goat anti-rabbit antibody (Abcam; ab150078) solution prepared in 1% (*w*/*v*) bovine serum albumin in DPBS (1:1600). The cell nuclei were stained (15 min, RT) with Hoechst 33242 Staining Dye Solution (diluted in PBS to 1 µg/mL; Abcam). The ProLong^TM^ Diamond Antifade Mountant (Thermo Fisher Scientific Inc.) was used for mounting the samples (24 h, RT, darkness). The stained samples on the microscope slides were analyzed via fluorescent microscopy using ImageJ Software version 1.54p (National Institute of Health, Bethesda, MD, USA), as described previously [81].

### 4.22. Experimental Setup and Statistical Analyses

The Mann–Whitney test was employed to compare the WT and KD groups due to donor variability and treatment effects suggesting a non-normal data distribution. The Kruskal–Wallis test was used to compare the medians of three or more groups, as done in the osteogenic differentiation experiments, followed by Dunn’s correction for multiple comparisons to ensure accurate significance levels. Analyses were conducted using Prism (Version 10.1.0, GraphPad Software, Boston, MA, USA), with a significance threshold set at *p* < 0.05. Data are expressed as mean ± standard error of the mean (SEM; the experiments included specified replication levels: the number of independent mixed donor-derived cultures (n, primary cell cultures), biological replicates per donor (n, biological per primary cell culture), and technical measurements per replicate (n, technical per biological replicate).

## 5. Conclusions

This study provides critical insights into the role of XYLT1 in hMSC differentiation, particularly highlighting its influence on chondrogenic and adipogenic pathways. Here, we first shed light on the cellular processes in DBQD2, which provide an explanation for the abnormal bone homeostasis. Our findings reveal that while osteogenic differentiation remains largely unaffected, the absence of XYLT1 drives premature chondrocyte hypertrophy, alongside reduced leptin expression in adipogenesis. These results underscore the importance of XYLT1 in regulating cellular functions within the bone marrow niche and suggest potential avenues for therapeutic intervention—not only concerning DBQD2 but also other skeletal disorders. The differential impacts of XYLT1 observed underscore its complex role in maintaining the balance between various cell types involved in bone homeostasis. By advancing our understanding of these mechanisms, this research lays the groundwork for future studies aimed at uncovering novel strategies to enhance bone health and address skeletal disorders.

## Figures and Tables

**Figure 1 ijms-26-07363-f001:**
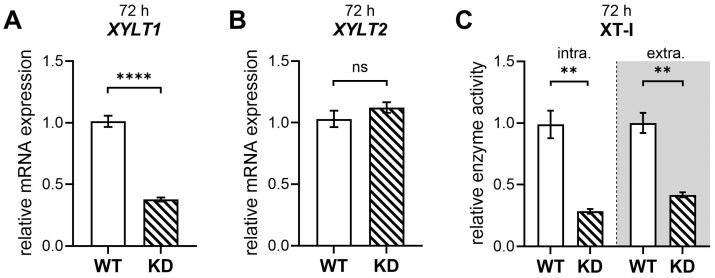
CRISPR-Cas9-mediated *XYLT1* targeting of hMSCs showed diminished *XYLT1* mRNA expression and XT-I activity. Primary hMSCs were transfected either without (WT) or with (KD) the RNP complex and cultured until reaching 90% confluence. The cells were subcultured at a density of 50 cells/mm^2^ and grown for 72 h before RNA extraction and complementary DNA synthesis for relative quantitative real-time polymerase chain reaction (qRT-PCR) analysis of (**A**) *XYLT1* and (**B**) *XYLT2* gene expression. (**C**) Intracellular (intra.) and extracellular (extra.) XT-I activities were determined using an XT-I isoform-selective UHPLC-ESI-MS/MS-based activity assay and normalized to the total protein content of the lysates. Data are means ± SEM from n = 2 primary cell cultures, n = 3 biological replicates per primary cell culture, and n = 3 technical replicates per biological replicate, normalized to their respective controls. Significance levels: not significant (ns), *p* < 0.01 (**), *p* < 0.0001 (****).

**Figure 2 ijms-26-07363-f002:**
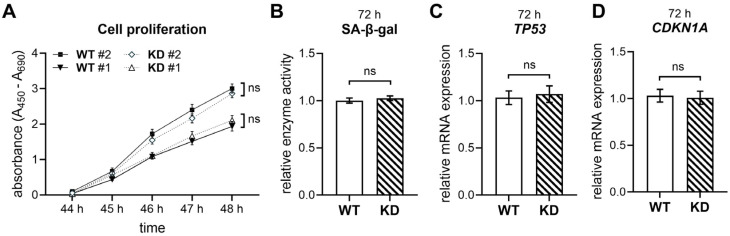
Cellular proliferation and senescence in hMSCs are unaffected by *XYLT1* deficiency. Primary hMSCs were transfected either without (WT) or with (KD) the RNP complex and cultured until reaching 90% confluence. The cells were subcultured at a density of 50 cells/mm^2^ and grown for 48 h to assess cellular proliferation and for 72 h to detect premature senescence. (**A**) Cellular proliferation was evaluated by adding the tetrazolium salt WST-1 to the cell culture supernatants at the 44 h time point, with measurements taken at 0, 1, 2, 3, and 4 h post-supplementation. The results are presented as means ± SEM from n = 2 primary cell cultures and n = 6 biological replicates per primary cell culture. Results are shown for each primary cell culture individually (WT #1 and WT #2 as well as KD #1 and KD #2). (**B**) The SA-β-gal assay was conducted after 72 h of cell regrowth, with results normalized to total protein content. (**C**) *TP53* (p53) and (**D**) *CDKN1A* (p21) mRNA expression levels were determined after 72 h using qRT-PCR. The data are means ± SEM from n *=* 2 primary cell cultures, n = 3 biological replicates per primary cell culture, and n = 3 technical replicates per biological replicate, normalized to their respective controls. Mann–Whitney test significance level: not significant (ns).

**Figure 3 ijms-26-07363-f003:**
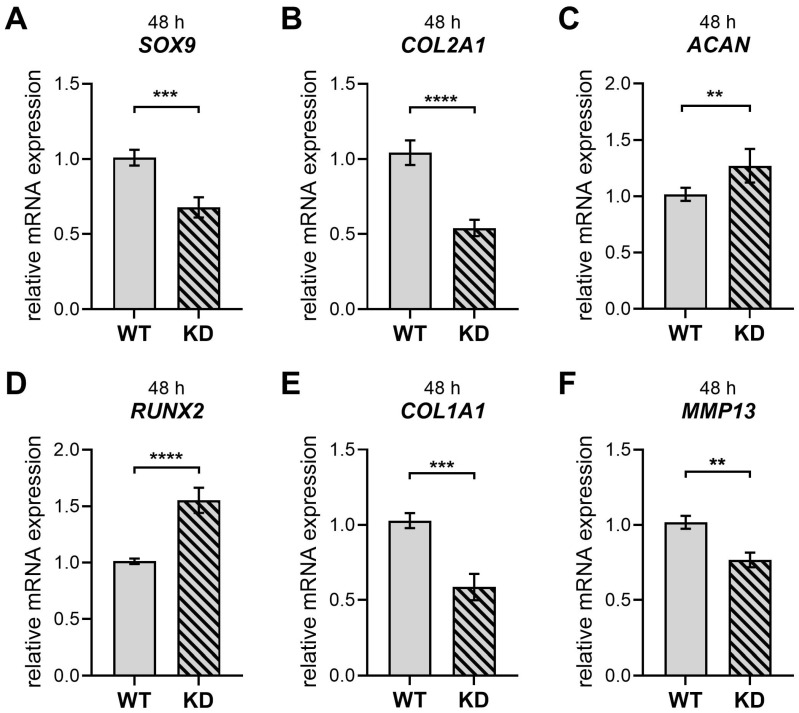
*XYLT1* deficiency leads to an aberrant expression of chondrogenically differentiated hMSCs. The hMSCs were cultured to 90% confluence following transfection without (WT) or with (KD) the RNP complex. Chondrogenic differentiation of hMSCs was performed using a high-density pellet culture with 250,000 cells maintained in chondrogenic differentiation medium. Gene expression analysis upon early chondrogenesis was performed at 48 h by determining the mRNA levels of (**A**) *SOX9*, (**B**) *COL1A2*, (**C**) *ACAN*, (**D**) *RUNX2*, (**E**) *COL1A1*, and (**F**) *MMP13* via qRT-PCR. Data are means ± SEM from n = 2 primary cell cultures, n = 3 biological replicates per primary cell culture, and n = 3 technical replicates per biological replicate, normalized to the WT sample. Mann–Whitney test significance levels: *p* < 0.01 (**), *p* < 0.001 (***), and *p* < 0.0001 (****).

**Figure 4 ijms-26-07363-f004:**
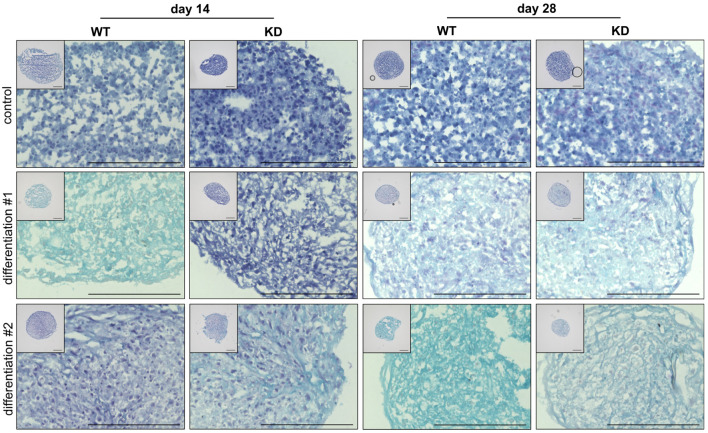
*XYLT1* deficiency alters the ECM deposition/PG content of chondrogenically differentiated hMSCs. The hMSCs were cultured to 90% confluence following transfection with (KD) or without (WT) RNP complex. A total of 250,000 cells were formed into spheroids and cultured in either hMSC standard medium (control) or chondrogenic differentiation medium (differentiation). The pellet cultures were fixed after 14 or 28 days and incubated in a 15% sucrose solution, and 5 μm cryosections were prepared. Cryosections were stained with a 1% alcian blue solution to visualize acidic PG and glycoproteins. Cell nuclei were counterstained blue to purple using Gill’s III hematoxylin solution. Representative images are shown from n = 2 primary cell cultures (indicated with #1, #2), n = 3 biological replicates per primary cell culture, and n = 3 technical replicates per biological replicate. Scale bar: 250 μm.

**Figure 5 ijms-26-07363-f005:**
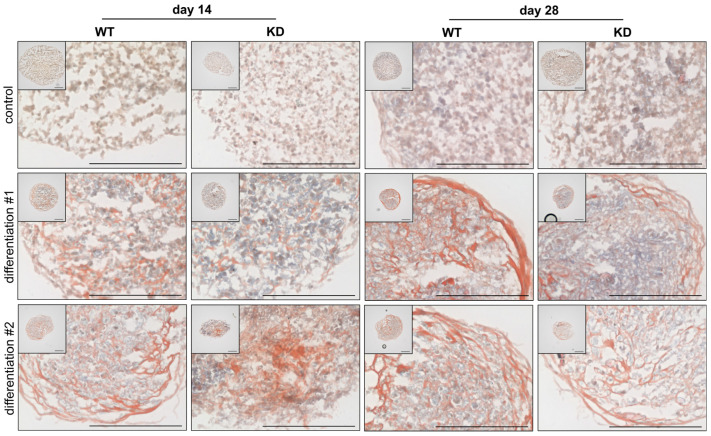
*XYLT1* deficiency affects the collagen deposition of chondrogenically differentiated hMSCs. The hMSCs were cultured to 90% confluence following transfection with (KD) or without (WT) RNP complex. A total of 250,000 cells were pelleted and cultured in either hMSC standard (control) or chondrogenic differentiation (differentiation) medium. The pellet cultures were fixed after 14 or 28 days and incubated in a 15% sucrose solution, and 5 μm cryosections were prepared. Cryosections were stained with Picro-Sirius red solution to visualize collagenous structures in orange-red. Cell nuclei were stained brown-black with acid-resistant Weigert’s iron hematoxylin solution. Representative images are shown from n = 2 primary cell cultures, n = 3 biological replicates per primary cell culture, and n = 3 technical replicates per biological replicate. Scale bar: 250 μm.

**Figure 6 ijms-26-07363-f006:**
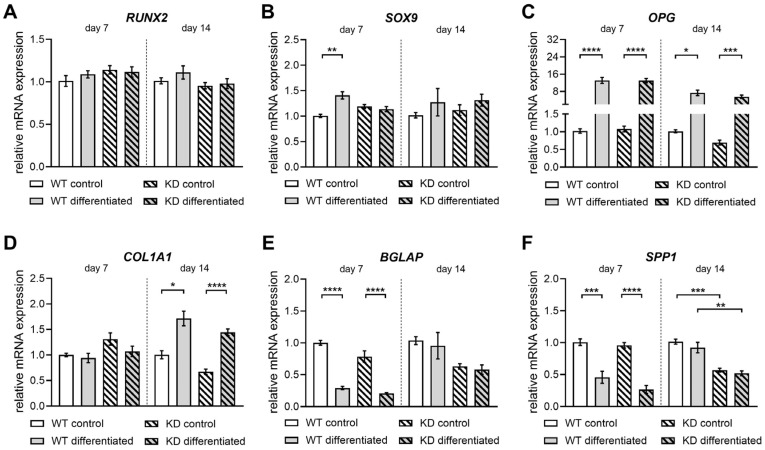
Analysis of osteogenic differentiation in WT and KD hMSCs via qRT-PCR at days 7 and 14. Primary hMSCs were transfected either without (WT) or with (KD) the RNP complex and cultured until reaching 90% confluence. The cells were subcultured at a density of 100 cells/mm^2^, grown until 100% confluence, and then cultivated for 7 or 14 days in either hMSC standard medium (control) or osteogenic differentiation medium (differentiated). The mRNA expression levels of the (**A**) *RUNX2*, (**B**) *SOX9*, (**C**) *OPG*, (**D**) *COL1A1*, (**E**) *BGLAP* (osteocalcin), and (**F**) *SPP1* (OPN) genes were determined at both day 7 and 14 of osteogenic differentiation. Data are presented as means ± SEM from n = 2 primary cell cultures, n = 3 biological replicates per primary cell culture, and n = 3 technical replicates per biological replicate, with normalization to the respective WT control. Kruskal–Wallis significance levels: *p* < 0.05 (*), *p* < 0.01 (**), *p* < 0.001 (***), and *p* < 0.0001 (****).

**Figure 7 ijms-26-07363-f007:**
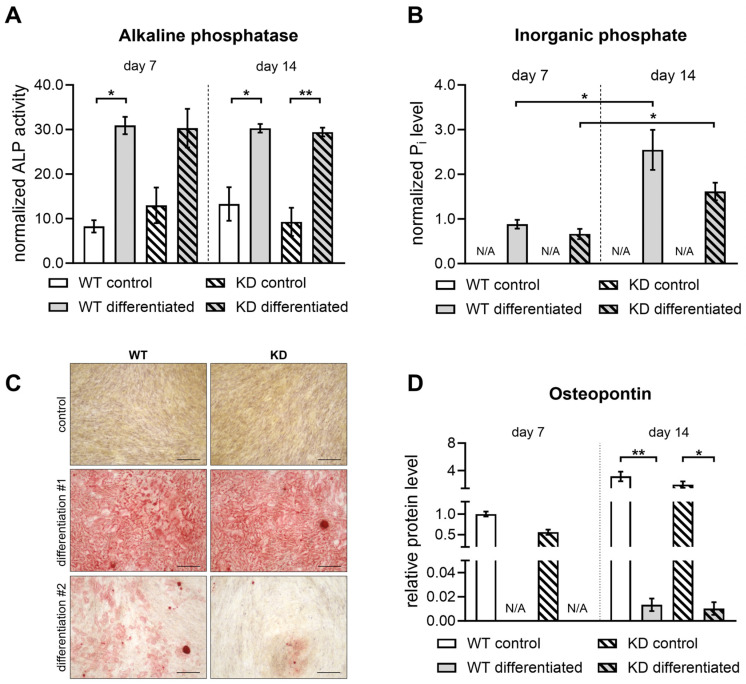
Comparison of the mineralization process in WT and KD hMSCs during osteogenesis. The hMSCs were cultured post-transfection without (WT) or with (KD) RNP complex until reaching 90% confluence. Cells were subcultured at a density of 100 cells/mm^2^ and, upon reaching 100% confluence, cultured for 7 or 14 days in either hMSC standard medium (control) or osteogenic differentiation medium (differentiated). (**A**) The ALP activity in cell lysates was quantified spectrophotometrically in technical duplicates. The ALP activities (mU/µg) were normalized to the protein content of cell lysates. (**B**) Free inorganic phosphate (P_i_) levels (µg/ng) from cell culture supernatants were measured using the ARCHITECT c8000 clinical chemistry analyzer and normalized to the DNA content of the respective cell lysates. (**C**) Calcium deposits were visualized by Alizarin Red S staining using two biological replicates per condition and time point. Representative images of the WT and KD hMSCs cultures on day 7 are shown. Scale bar: 100 µm. (**D**) Osteopontin protein levels in conditioned media were quantified by ELISA, with differentiation controls diluted 1:10 and protein levels normalized to the respective DNA content of the cell lysates. Data are means ± SEM from n = 2 primary cell cultures, n = 3 biological replicates per primary cell culture, and n = 3 technical replicates per biological replicate. Kruskal–Wallis significance levels: *p* < 0.05 (*), *p* < 0.01 (**). N/A = not available.

**Figure 8 ijms-26-07363-f008:**
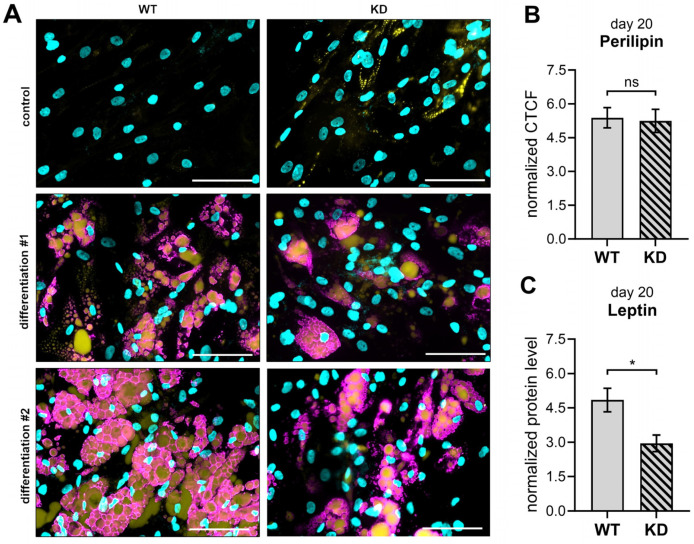
Quantification of perilipin 1 and leptin protein expression in WT and KD adipogenically differentiated hMSC cultures. The hMSCs were cultured to 90% confluence following transfection without (WT) or with (KD) the RNP complex. Cells were subcultured at a density of 210 cells/mm^2^ in standard medium until reaching full confluency. Adipogenic differentiation was conducted over 20 days with cycles of induction and maintenance media, followed by 7 days in maintenance media. Controls remained in maintenance media throughout. (**A**) Cells were fixed for the fluorescence detection of perilipin 1 (magenta) with BODIPY-staining lipid droplets (yellow) and DAPI counterstaining nuclei (cyan). Scale bar = 100 μm. (**B**) Perilipin 1 expression (CTCF) was quantified using ImageJ by analyzing eight images per biological replicate, normalized to total cell number per image. (**C**) Leptin protein levels (pg/ng) in 1:10 diluted cell lysates were measured via ELISA in duplicate, normalized to total DNA. Data are means ± SEM from n = 2 primary cell cultures and n = 3 biological replicates per primary cell culture. Mann–Whitney test significance levels: not significant (ns), *p* < 0.05 (*).

**Table 1 ijms-26-07363-t001:** Donor characteristics of hMSCs used in this study.

Lot Number	Donor Age	Sex	Designation	Race	Tissue/Localization	Surface Markers *
465Z016	63	male	#1	Caucasian	Human bone marrow/femoral head	CD73^+^, CD90^+^, CD105^+^, CD14^−^, CD34^−^, CD45^−^, CD19^−^, HLA-DR^−^
467Z023.5	68					
466Z020	44	female	#2			
451Z012.3	66					

* ≥90% positive for positive markers, ≤10% positive for negative markers.

**Table 2 ijms-26-07363-t002:** Primer systems used for qualitative real-time polymerase chain reaction analysis (T_A_:PCR annealing temperature).

Gene	5′ to 3′ Sequence	T_A_ [°C]	Efficiency
*ACAN*	CACCCCATGCAATTTGAG GCCACTGTGCCCTTTTTA	63	2.00
*B2M*	TGTGCTCGCGCTACTCTCTCTT CGGATGGATGAAACCCAGACA	63	1.87
*BGLAP*	CGCCTGGGTCTCTTCACTAC CTCACACTCCTCGCCCTATT	66	1.84
*CDKN1A*	GCTTCATGCCAGCTAACTTCC CCCTTCAAAGTGCCATCTGT	66	2.00
*COL1A1*	GATGTGCCACTCTGACT GGGTTCTTGCTGATG	63	1.74
*COL2A1*	GGGCTCCCGCAAGAA GCAGGCGTAGGAAGGTCA	63	1.84
*GAPDH*	AGGTCGGAGTCAACGGAT TCCTGGAAGATGGTGATG	59	1.83
*HMBS*	CTGCCAGAGAAGAGTGTG AGCTGTTGCCAGGATGAT	63	1.92
*MMP13*	AGCTGGACTCATTGTCGGGC AGGTAGCGCTCTGCAAACTGG	63	1.75
*OPG*	CGGGAAAGAAAGTGGGAGCAG CTTCAAGGTGTCTTGGTCGCCAT	63	1.99
*RPL13A*	CGGAAGGTGGTGGTCGTA CTCGGGAAGGGTTGGTGT	63	1.87
*RUNX2*	AGAAGGCACAGACAGAAGCTTGA AGGAATGCGCCCTAAATCACT	61	1.82
*SOX9*	TACCCGCACTTGCACAAC TCTCGCTCTCGTTCAGAAGTC	63	2.00
*SPP1*	TGATGACCATGTGGACAG ACCATTCAACTCCTCGCT	61	1.90
*TP53*	AGATAGCGATGGTCTGGC TTGGGCAGTGCTCGCTTAGT	63	2.00
*XYLT1*	GAAGCCGTGGTGAATCAG CGGTCAGCAAGGAAGTAG	63	2.00
*XYLT2*	ACACAGATGACCCGCTTGTGG TTGGTGACCCGCAGGTTGTTG	63	1.95
*YWHAZ*	CTCCCGTTTCCGAGCCATAA AAGATGACCTACGGGCTCCT	63	2.00

## Data Availability

The data presented in this study are available on request from the corresponding author.

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
