# Peer review of "XYLT1 Deficiency of Human Mesenchymal Stem Cells: Impact on Osteogenic, Chondrogenic, and Adipogenic Differentiation"

_ijms, 2025, doi:10.3390/ijms26157363_

Round 1
Reviewer 1 Report
Comments and Suggestions for Authors
The manuscript by Thanh-Diep Ly et al claims that XT-I’s pivotal role in regulating differentiation pathways within the bone marrow niche, influencing cellular functions critical for skeletal health. However, the manuscript couldn’t be accepted. The comments and questions are followed.
- My main concern is related to the number of data, the manuscript showed that biological and technical replicates per experiment (n = 2) in all figures. The Mat & Met and figure legends section should be more precise and detailed (n). T To make sure the accurate conclusion, The number of experiments is at least 3 in every group.
- Cellular proliferation in Figure2A was detected every 1 hour, why so short? In common, at least 8-12 h. What’s difference of WT1 and WT2?
- RUNX2 and COL1A1 aren’t chondrogenic differentiation markers in Figure 3, which is osteogenic differentiation markers.
- Author showed so much data, however, I can’t find the main conclusion and significance.
Reviewer 2 Report
Comments and Suggestions for Authors
Dear authors,
congratulations for your work.
The manuscript is well written, fluent and has a good scientific impact.
It also has a good technological influence and is constructed according to a rational scientific workflow.
I suggest only minors modifications:
- Lane 140: typo error.
- Reduce and summarize M&M and Discussion sections.
- Change the graph in Figure 1C, respect to the figure1A and 1B
- Lane 587: not only vital bone cells. There are several cell types derived from hMSC
- Explain better how did you use the primary cells. Are donors cells used as a POOLED batch?
- Are this POOL the same for each experiments?
- There is just one issue with the paragraph title " Diminished XYLT1 expression of hMSCs does not critically impair their differentiation into 729 osteoblasts " and the overall title "XYLT1 deficiency of human mesenchymal stem cells impairs adult bone and cartilage homeostasis”
- 5. XYLT1-deficient adipogenic differentiated hMSCs possess a diminished leptin expression”. Please insert this aspect in the overall title.
